# Prognostic Role of CD200 in Acute Lymphoblastic Leukemia Patients

**DOI:** 10.3390/diagnostics13020325

**Published:** 2023-01-16

**Authors:** Mohamed Khalil, Nahla Elsharkawy, Mona Mohsen Elmawardy, Mahmoud Aly Ayoub

**Affiliations:** Department of Clinical and Chemical Pathology, National Cancer Institute, Cairo University, Cairo 11796, Egypt

**Keywords:** acute lymphocytic leukemia, CD200, minimal residual disease

## Abstract

Background: Overexpression of CD200 in ALL patients indicates that it may be useful in the characterization of leukemia initiating cells (LIC). We aim at investigating the expression pattern of CD200 on leukemic B cells and the correlation of CD200 expression with various clinical and laboratory findings in 62 newly diagnosed acute lymphoblastic leukemia patients. Methods: All patients were subjected to full history taking, a thorough clinical examination, and laboratory investigations, which included complete blood count (CBC), BM aspiration, immunophenotyping of blast cells, and CD200 expression. Results: There is a higher statistically significant mean value of CD200 expression among the cases (66.15 ± 23.08) than the control group (0.37 ± 0.2) (*p* value ≤ 0.001). CD200 expression shows a significant correlation with total leucocytic count and hemoglobin level (*p* = 0.001, 0.03, respectively). Conclusions: This study showed that CD200 expression was expressed in 100% of the patients. Correlations between CD200 expression and different laboratory data of patients revealed that there was an impact of CD200 on different diagnostic findings. After the follow-up of the patients, we found that the use of PRISM function of the software could add value to the detection of minimal residual disease.

## 1. Introduction

Acute lymphoblastic leukemia (ALL) comprises about 30% of all malignancies in children. ALL is most common in preschool children. Another incidence peak is seen in adults aged over 50 years. The identification of new prognostic factors has provided a means of stratifying patients into different risk groups and “tailoring” treatment accordingly [1]. The current challenge is identifying such prognostic factors at diagnosis, which determine high-risk ALL patients to assign a more aggressive therapy protocol to improve their outcome. On the other hand, this might avoid the more severe side effects of enhanced therapy in patients who can be treated with standard-intensity treatment [2].

Minimal residual disease (MRD) is an important predictor of relapse in acute lymphoblastic leukemia (ALL). It is used as a criterion for risk stratification in many current studies, but its relationship to other prognostic variables has not been fully researched [3]. Two well-established techniques, flow cytometry (FCM) and polymerase chain reaction (PCR), can detect leukemic cells with a sensitivity of 0.01% [4]. Flow cytometry is more patient-specific, quicker, often less costly, and more likely to produce useful data than molecular approaches [3].

CD200 (cluster of differentiation 200) (OX-2 antigen) is a type I immunoglobulin superfamily membrane glycoprotein. It is expressed in multiple cell types, including B cells, a subset of T cells, dendritic cells, endothelial cells, and in the peripheral and central nervous systems. CD200 interacts with CD200R, an immunoglobulin superfamily inhibitory receptor expressed primarily on myeloid/monocyte lineage cells and a subset of T cells. CD200R suppresses monocyte and T cell-mediated immune responses [5]. In ALL, Coustan-Smith et al. found that CD200 was overexpressed in both high hyper-diploidy and ETV6-RUNX1 subtypes (ets variant gene 6-Runt-related transcription factor 1) [6].

This study aims to investigate the expression pattern of CD200 on leukemic B cells and to determine whether CD200 could emerge as a new tool for the detection of B-ALL cases and the incorporation of CD200 into routine MRD panels. We will correlate CD200 expression with various clinical and laboratory findings in newly diagnosed acute lymphoblastic leukemia patients.

## 2. Materials and Methods

The present study was conducted on 62 newly diagnosed acute lymphoblastic leukemia patients who attended the adult and pediatric hematology/oncology clinic of the National Cancer Institute. Patients were followed up throughout the period of the study (one year). Thirty age- and sex-matched healthy subjects were taken as a control group; they were 21 males and 9 females, aged between 5 and 50 years.

The pediatric protocol was based on total XV St. Jude protocol, while adults received the Dana Farber Cancer Institute (DFCI) ALL/LBL protocol [7].

The samples used were either peripheral blood or bone marrow samples for both cases and controls. BM samples were used in the follow-up of the cases to assess MRD.

All patients were subjected to full history taking, a thorough clinical examination, and laboratory investigations, which included complete blood count (CBC), BM aspiration, immunophenotyping of blast cells, and CD200 expression.

### 2.1. Immunophenotyping

It was performed by using Navious EPICS XL (Coulter Corporation, Hialeah, FL, USA) flow cytometry and cytogenetic analysis. CD200 was assessed using flow cytometry on either PB or BM samples of the B-ALL cases as well as PB or BM samples of the control group according to the method described by Cox et al. [8].

DNA index (DI) analysis and karyotyping were performed on the patients to detect hypoploidy. On days 15 and 42 following therapy initiation, remission achievement was assessed. The minimal residual disease is the accumulation of at least 10 clustered events out of 100,000 events that exhibit lymphoid-scattering qualities and leukemia-associated immunophenotypic traits [9].

Morphological remission was considered when blasts in the bone marrow <5% [10]. However, immunological remission was considered when minimal residual disease by flow cytometry <0.01%. To determine whether the initial percent of expression of CD200 could discriminate prognosis, the correlation of CD200 expression with total leucocytic count and hemoglobin level was used to calculate the best cutoff value for CD200 expression in B-ALL. The good prognosis was defined as (Total leucocytic count <50,000/mm^3^ and Hemoglobin < 10 gm% [11].

The Navios Cytometer software PRISM is used to analyze multicolor immunofluorescence samples. PRISM allows you to display percentages for all phenotypic populations in a single plot. It is software-derived and can be acquired in either run-time or list mode [12].

### 2.2. Statistical Analysis

For data analysis, the SPSS software (version 17.0) was utilized. The mean and standard deviation were quantitative data estimations. The Mann-Whitney test compared the means of two independent groups, and the Kruskal-Wallis test compared the means of more than two groups. Chi-square and Fischer exact were percentage independence tests. The Kaplan-Meier technique was used to get the total log-rank of survival curves. Correlation analysis was used to determine the degree of correlation between various numerical variables, including research markers. The cut-off level for distinguishing leukemic patients from normal ones was determined using receiver operator characteristic curve (ROC) analysis. A *p*-value below 0.05 was considered significant.

## 3. Results

### 3.1. Subsection

#### 3.1.1. The Clinical and Laboratory Characteristics

The present study was conducted on 62 newly diagnosed B-ALL patients attending the adult and pediatric hematology/oncology clinic of the National Cancer Institute. They were 40/62 males (64.5%) and 22/62 females (35.5%). The children were 48/62 (77.5%) and their ages ranged from 1 to 18 years, with a mean value of 6.729 ± 5.188 years, and the adults were 14/62 (22.5%) and their ages ranged from 20 to 79 years, with a mean value of 43.571 ± 17.99 years. Out of these 35 cases analyzed, 21 were pediatric (60%) and 14 were adults (40%).

All patients were followed up throughout the study period (one year). Thirty age- and sex-matched healthy subjects were taken as a control group. They were 21 males (70%) and 9 females (30%). The children were 17/30 and the adults were 13/30. Their age ranged from 5 to 50 years, with a mean value of 20.33 ± 12.35.

Platelets, hemoglobin level, and total leucocytic count are shown in Table 1. There was no significant correlation between CD200 expression and the clinical outcome; 35/62 (56.5%) were in complete remission (CR), 9/62 (14.5%) lost follow-up, and 18/62 (29%) died.

#### 3.1.2. Immunophenotyping

Out of 62 B-ALL patients, 22 (35.5%) were of the common ALL (c-ALL or CD10 positive) phenotype. Pre-B phenotype (CD10+, expression of cytoplasmic IgM (Cyto µ) heavy chain) represented 62.9% of patients (39 patients). One patient was of the Pro-B phenotype (CD10 negative), which represents 1.6%. One case (1.6%) showed aberrant expression of myeloid antigens, CD33, and it was of the c-ALL phenotype.

DNA index (DI) and karyotyping

DNA index (DI) showed that 10 (16.9%) had a high DNA index (>1.16), which indicates a hyper-diploid karyotype. No patients had a low DNA index (<0.95), which is considered hypodiploidy. The rest of the patients (49 patients, 83.1%) had a diploid DNA index between 0.95 and 1.16.

Karyotyping was performed on thirty-five patients with a mean value of normal cases 12/35 (28.53 ± 5) and abnormal cases 23/35 (12.26 ± 3.5). Out of these 35 cases analyzed, 21 were pediatric (60%) and 14 were adults (40%). Results showed a statistically non-significant difference in CD200 expression between normal cases (28.53 ± 5) and abnormal cases (12.26 ± 3.5) (*p*-value = 0.389). When comparing the different karyotyping categories as regards CD200 expression, results showed a statistically non-significant difference in CD200 expression within abnormal karyotypes: hyper-diploid cases 13/23 (56.7 ± 33) and other different abnormal cases (10/23 (66.8 ± 21.8); *p*-value = 0.389), of which five patients were positive for t (9;22) and two patients were positive for t (12;21). 

When comparing the different cytogenetic categories as regards CD200 expression, results showed a statistically significant difference in CD200 expression between normal cases (28.53 ± 5) and abnormal cases (12.26 ± 3.5) (*p*-value = 0.005) (Table 2).

When comparing the abnormal karyotyping categories as regards CD200 expression, results showed a statistically non-significant difference in CD200 expression between hyperdiploid cases 13/23 (56.7 ± 33) and other abnormal cases 10/23 (66.8 ± 21.8) (*p*-value = 0.389) (Table 3).

##### CD200 Expression in the B-ALL Group and the Control Group

Analysis of CD200 expression was performed for 62 patients and 30 control cases. In the B-ALL group, CD200 expression ranged from 2.4–98 (%), with a mean value of 66.15 ± 23.08. In the control group, CD200 expression ranged from 0–0.585 (%), with a mean value of 0.37 ± 0.2. There was a high statistically significant difference between the B-ALL group and the control group, with a high expression in the B-ALL group (*p* < 0.001).

##### Relation between CD200 Expression and Different Parameters

There was no statistically significant difference regarding sex (*p* = 0.91) and age (*p* = 0.49), hepatomegaly, splenomegaly, and lymphadenopathy in CD200 expression compared to those without organomegaly or lymphadenopathy (*p* = 0.85, 0.31, and 0.80, respectively). Considering the laboratory data, there was a statistically significant difference between CD200 expression, total leucocytic count, and hemoglobin level (*p* = 0.001 and 0.03, respectively). There was no statistically significant difference between CD200 expression, platelet count, and bone marrow blasts (*p* = 0.11 and 0.219, respectively).

##### CD200 Expression and Other B-Cell Markers

Correlation studies between CD200 expression and other B-cell markers such as CD10, CD19, CD22, and Cyto µ showed no statistically significant correlation except for CD10 and CD19, which were statistically significant (*p* = 0.002, 0.009, 0.294, and 0.91, respectively). (Figure 1). There was no statistically significant difference in CD200 expression between the different subtypes (*p*-value = 0.4). The comparison was not performed on Pro B and C-ALL with aberrant CD33, as there was only one patient in each of them. There was no significant correlation between CD200 initial expression and the DNA index, with a *p*-value of 0.496.

##### CD200 Cutoff Value

The receiver operating characteristic curve (ROC curve) for the determination of the cutoff of CD200 was performed with 100% sensitivity and 100% specificity. The area under the ROC curve = 1.0. CD200 > 1% was considered positive for B-ALL cases as compared to the control group. The best cutoff value was 60%. The smallest cutoff value is the minimum observed test value minus one, and the largest is the maximum value minus one. All the other cutoff values are the averages of two consecutively ordered observed test values. Therefore, we have chosen 60% as our cutoff value.

#### 3.1.3. Clinical Outcome and Follow-Up

Thirty-nine patients were followed up for detection of minimal residual disease (MRD) on day 15 of induction therapy and 28 patients on day 42. Results showed that at day 15, 26/39 patients (66.7%) achieved immunological remission (negative MRD ≤ 0.01%), and all of them (26/26, 100%) were in morphological remission (blast < 5%), while 13/39 patients (33.3%) did not achieve immunological remission (positive MRD > 0.01%). In the meantime, 7/13 (53.8%) were in morphological remission (Table 4). At day 42, 23/28 patients (82%) achieved immunological remission, and all (23/23,100%) were in morphological remission, while 5/28 patients (18%) did not achieve immunological remission, and 5/5 (100%) were in morphological remission (Table 4; (Figure 2). The correlation between initial CD200 expression and MRD D15 and MRD D42 showed no statistically significant correlation with a *p*-value of 0.393 and 0.317, respectively.

Patients were categorized into two groups according to the initial expression of CD200 (≤60% and >60%). Results showed that there was no statistically significant difference between the two groups at day 15, where 72.7% of patients who had initial CD200 expression ≤60% achieved immunologic remission, while 64.3% of those who had initial CD200 expression >60% achieved immunologic remission (*p*-value = 0.72) (Table 5). At day 42, results showed that there was no statistically significant difference between the two groups, where 100% of patients who had initial CD200 expression ≤60% achieved immunologic remission, while 77.3% of those who had initial CD200 expression >60% achieved immunologic remission (*p*-value = 0.55) (Table 5).

MRD D15 was compared in twenty of the cases using the combination of CD38 or CD58/CD200/CD19/CD10/CD34 (PRISM), where 100% of the cases who were negative by PRISM achieved immunological remission, while 22.2% of the cases who were positive by PRISM achieved immunological remission (Table 6). Eleven of the cases were negative by PRISM and negative by MRD, while zero of the cases were negative by PRISM and positive by MRD.

Two of the cases were positive by PRISM and negative by MRD (which indicates that the PRISM function of the software may add value to the MRD), and seven of the cases were positive by PRISM and positive by MRD.

##### Overall Survival

Regarding the relationship between death and immunologic remission, 88.5% of patients who achieved immunologic remission at day 15 survived, while 84.6% of those who did not achieve immunologic remission survived, but there was no statistically significant difference (*p* = 1.0; Table 7). The correlation between initial CD200 expression and death showed no statistically significant correlation with a *p*-value of 0.86. (Figure 3)

One-year overall survival (OS) probability was evaluated by log-rank analysis of Kaplan-Meier plots comparing the B-ALL patient group with initial CD200 expression ≤60% to those with initial CD200 expression of >60%. Results revealed that there was no significant influence of CD200 expression on the overall survival probability (*p* = 0.375). The one-year probability of overall survival for the B-ALL patient group with CD200 expression ≤60% was 78.9%, compared to 67.4% for those with CD200 expression >60% (Figure 4).

## 4. Discussion

Acute lymphoblastic leukemia (ALL) is the most common type, accounting for about 80% of leukemia incidence among children below the age of 14 years [13]. Regarding the correlation of CD200 expressionwith the demographic data, CD200 expression showed no significant difference when comparing the expression in pediatrics and adults (*p* value = 0.49). Correlation studies showed no significant associations between CD200 expression and different clinical findings; when compared to different laboratory data, there was no significant correlation except for total leucocytic count and hemoglobin level (*p* = 0.001 and 0.03, respectively). In the current study, a white blood cell (WBC) count at diagnosis >50.0 × 10^9^ cells/L was registered in 21% of cases, matching Brazilian and Canadian studies that reported WBC counts >50.0 × 10^9^ cells/L in 21% and 20% of cases, respectively [14,15].

For the diagnosis of B-ALL, a broad leukemia marker panel was applied, including common progenitor markers (CD34, HLA-DR), panleucocytic marker (CD45), B-cell markers (CD10, CD19, CD22, cyt IgM, cyt CD79α, Kappa, Lambda), T-cell markers (CD2, CD3, CD4, CD5, CD7, and CD8), and myeloid markers (CD13, CD14, and CD33).

In this study, DNA index analysis showed that 10 (16.9%) had a high DNA index (>1.16), which indicates a hyperdiploid karyotype. No patients had a low DNA index (<0.95), which is considered hypodiploidy. The rest of the patients (49 patients, 83.1%) had a diploid DNA index of 0.95–1.16. The correlation was conducted between CD200 initial expression and DNA index, which reveals no significant correlation. These results are in concordance with Lustaza et al. [14] DI >1.16 was associated with a favorable prognosis, and at the end of induction, all patients with DI >1.16 were alive and in complete remission with no early recurrence. Arico et al., reported that cases with DI >1.16 were 23% and found a strong association between hyperdiploidy and favorable prognostic factors such as age and between one and five and WBC count at diagnosis <20.0 × 10^9^ cells/L [14,16].

In the present work, CD200 expression was statistically significantly higher in B-ALL cases compared to healthy subjects. CD200 expression in B-ALL cases ranged from 2.4–98 (%), with a mean value of 66.15 ± 23.08, which is higher than the control group, where CD200 expression ranged from 0–0.585 (%), with a mean value of 0.37 ± 0.2. Similar results were obtained by Cox et al., who reported that CD200 was overexpressed in B-ALL cases compared to cord blood (CB) samples (54.6% ± 8.1% vs. 0.07 ± 0.09%, respectively, *p* = 0.0002) [8]. CD200 was expressed in 95% of cases in a study conducted by Alapat et al., and Dorfman et al., reported that CD200 was expressed in 100% of cases [17,18]. In a work conducted by Coustan-Smith et al., CD200 was overexpressed in both hyperdiploidy and t (12;21) groups [6]. Aref et al., found that 28/43 (or 65%) of B-ALL cases exhibited CD200, 5/43 (11.6%) expressed CD56, and only two patients (4.7%) expressed both CD200 and CD56. Patients who tested positive for CD200 and CD56 had a substantially lower platelet count and a worse propensity for inducing a remission response compared to negative patients (*p* = 0.01 for both). When comparing patients with CD200+ and CD56+ expression to those with CD200 and CD56 expression, the OS and DFS were considerably lower in the CD200+ and CD56+ cases. Positive expression of CD200 and/or CD56 upon diagnosis in B-ALL is related to a poor prognosis and may be indicative of biological aggressiveness [19]. This confirms our data regarding the prognostic value of CD200. Diamenti et al., used whole-genome microarrays and flow cytometric analysis to show that these compounds were overexpressed relative to normal controls. Leukemia engraftment in immune-deficient mice did not require the expression of CD58 or CD97, which were mostly co-expressed with CD19. In contrast, CD200 expression was significant for engraftment and repeated transplantation of cells in patients with a low risk of detectable residual disease (MRD). More so, the CD200+ LPCs may be targeted in vitro and in vivo with the monoclonal antibody TTI-CD200. Mice with an advanced illness were successfully treated, resulting in a dramatic decrease in disease severity and an increase in survival time. These results suggest that CD200 may be a promising therapeutic target for low-risk AML, with few of the off-tumor consequences that affect standard immunotherapies [20].

In this study, correlation of CD200 expression with total leucocytic count and hemoglobin level was used to calculate the best cutoff value for CD200 expression for the prediction of poor prognosis in B-ALL, and the best cutoff value was 60%. This was done to determine whether the initial percent of CD200 expression could predict the prognosis of ALL patients.

One-year overall survival (OS) probability was evaluated by log S–rank analysis of Kaplan-Meier plots comparing the B-ALL patient group with initial CD200 expression ≤60% to those with initial CD200 expression of >60%. The one-year probability of overall survival for the B-ALL patient group with CD200 expression ≤60% was 78.9% compared to 67.4% for those with CD200 expression >60%. Our results revealed that there was no significant influence of CD200 expression on the overall survival probability. The PRISM function of the software was used to estimate the percentage of cell populations expressing different antigen combinations. The PRISM processor represents data from 2, 3, 4, or 5 antibody combinations as 4, 8, 16, or 25 evenly spaced peaks in a single-parameter graph. Each peak represents an antibody combination, or phenotype. For example, one peak represents the cells that are negative for all parameters, the second peak represents the cells that are positive for the first parameter and negative for all other parameters, and the last peak represents the cells that are positive for all parameters. The height of the peak is proportional to the number of events belonging to the phenotype represented by the peak.

MRD D15 was compared in twenty of the cases using PRISM software with the combination of CD38 or CD58/CD200/CD19/CD10/CD34, where 100% of the cases that were negative by PRISM achieved immunological remission, while 22.2% of the cases that were positive by PRISM achieved immunological remission. Awad et al., used the PRISM function of the software in 84 newly diagnosed AML patients to characterize leukemia stem cells (LSCs) and to discriminate them from normal hematopoietic stem cells (HSCs) [21]. Coustan-Smith et al., reported that CD200 was stably expressed at the time of MRD, which denotes its potential value as an MRD marker and therapy target. Incorporating CD200 into routine MRD panels increases the sensitivity for detecting minimal residual clones and patients with suboptimal responses [6].

Correlation studies between CD200 expression and other B-cell markers such as CD10, CD19, CD22, and Cyto µ showed a statistically significant correlation for CD10 and CD19 only (*p* = 0.002 and 0.009, respectively). Moreover, when comparing the different B-ALL subtypes as regards CD200 expression, the results showed no statistically significant difference in CD200 expression between the different subtypes (*p* value = 0.4). The best cutoff value for CD200 (%) expression in the detection of B-ALL was found to be 1% with a sensitivity of 100% and specificity of 100%. On the other hand, the best cutoff value for CD200 (%) expression in predicting the prognosis was found to be 60%, which showed no significant correlation with MRD at days 15 and 42. MRD D15 was compared in twenty of the cases using the combination of CD38 or CD58/CD200/CD19/CD10/CD34/CD45 (PRISM), where 100% of the cases who were negative by PRISM achieved immunological remission, while 22.2% of the cases who were positive by PRISM achieved immunological remission with a significant *p*-value (*p* < 0.001).

More extensive studies should be conducted on this finding to confirm whether PRISM software could add value to MRD. Further studies on the prognostic value of CD200 expression should be performed on a wider scale of patients.

## 5. Conclusions

This study showed that CD200 expression was expressed in all the patients, while it was not expressed in the normal control group. CD200 was found to be stably expressed during MRD monitoring, and it increases MRD sensitivity by incorporating it in the MRD panel, indicating its potential role in MRD monitoring in ALL patients. These findings recommend the use of CD200 as a powerful tool for MRD detection in B-ALL cases. Correlations between CD200 expression and different laboratory data from patients revealed that there was an impact of that antigen on different diagnostic findings. With a follow-up of patients, we found that the use of the PRISM function of the software could add value to the detection of minimal residual disease. However, this finding was found in a small number of patients, so more studies should be performed on a larger scale of patients.

## Figures and Tables

**Figure 1 diagnostics-13-00325-f001:**
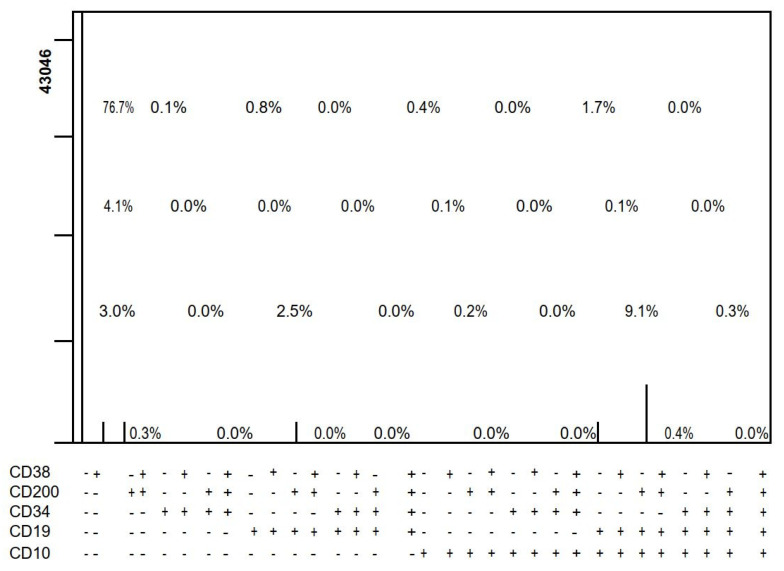
PRISM of one case at D14 with minimal residual disease.

**Figure 2 diagnostics-13-00325-f002:**
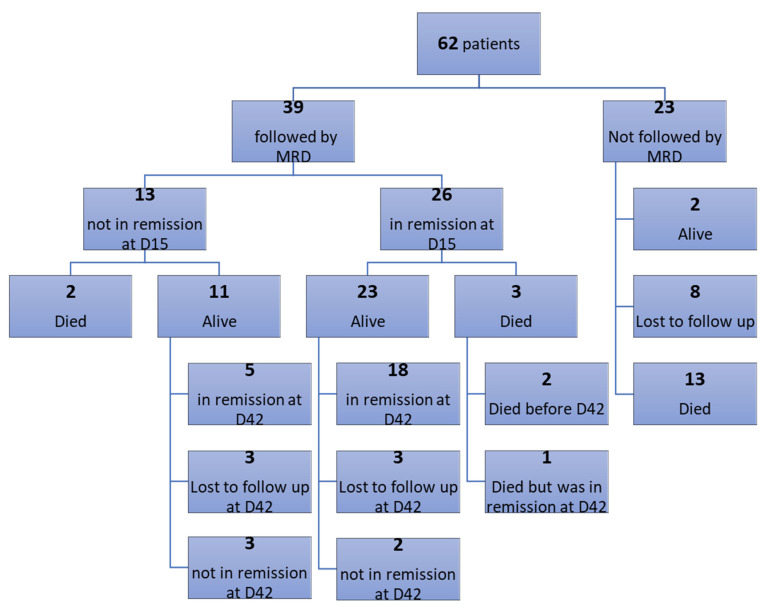
Algorithm follow-up of the cases.

**Figure 3 diagnostics-13-00325-f003:**
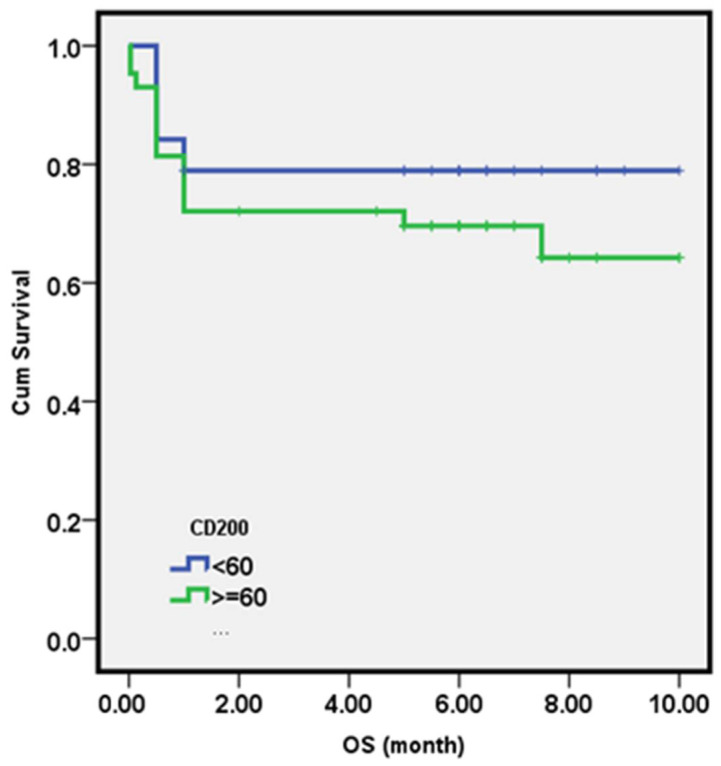
Overall survival probability of B-ALL patients with respect to CD200 expression.

**Figure 4 diagnostics-13-00325-f004:**
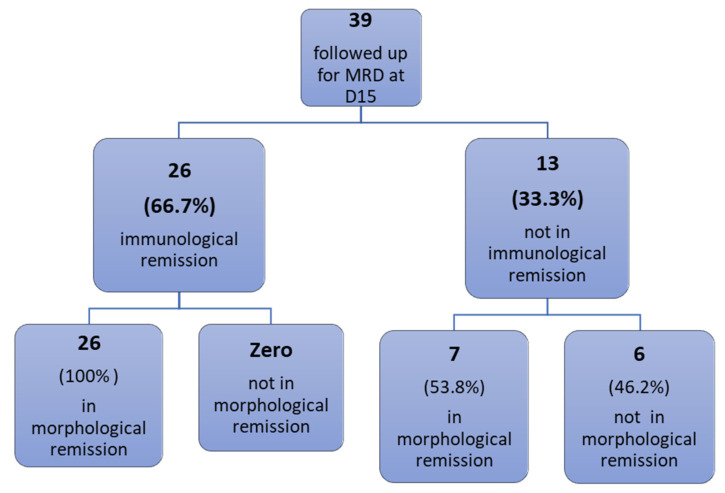
Distribution of patients with morphological remission versus immunological remission (negative MRD ≤ 0.01% by flowc ytometry) at day 15.

**Table 1 diagnostics-13-00325-t001:** Comparison between different laboratory data and CD200 expression with patient demographics.

		Number	Mean	Standard Deviation	*p*-Value *
Age	adults	14	43.571	17.99	
	children	48	6.279	5.188	
Sex (male to female ratio)		40/22	-	-	
TLC	<50,000	49	62.4776	23.88601	0.001
	≥50,000	13	80.0000	12.81926	
HB	<10	48	70.2792	20.52762	0.03
	≥10	14	52.0000	26.43715	
PLT	<50	30	61.2333	23.45161	0.11
	≥50	32	70.7625	22.11200	

* *p*-value ≤ 0.05 is considered statistically significant.

**Table 2 diagnostics-13-00325-t002:** Comparison between cytogenetic categories with regard to CD200 expression.

Cytogenetics	*N*	Mean ± SD	*p*-Value *
Normal	12	28.53 ± 5	0.005
Abnormal	23	12.26 ± 3.5	

* *p*-value ≤ 0.05 is considered statistically significant.

**Table 3 diagnostics-13-00325-t003:** Comparison between karyotyping categories with regard to CD200 expression.

Karyotyping	*N*	Mean ± SD	*p*-Value *
Hyper-diploid	13	56.7 ± 33	0.389
Other abnormal karyotype	10	66.8 ± 21.8	

* *p*-value ≤ 0.05 is considered statistically significant.

**Table 4 diagnostics-13-00325-t004:** Distribution of patients with morphological remission versus immunological remission on days 15 and 42.

	MRD D15		MRD D42	
Blasts (%)	negative (*N* = 26)	positive (*N* = 13)	negative (*N* = 23)	positive (*N* = 5)
<5	26 (100%)	7 (53.8%)	23 (100%)	5 (100%)
≥5	0 (0%)	6 (46.2%)	0 (0%)	0 (0%)

**Table 5 diagnostics-13-00325-t005:** Comparison of the immunologic remission status on days 15 and 42 between patients with initial CD200 expression levels (≤60% and >60%).

	CD200 %				*p*-Value *
	≤60%		>60%		
	Count	%	Count	%	
MRD D15					
−ve	8	72.7	18	64.3	0.72
+ve	3	27.3	10	35.7	
MRD D42					
−ve	6	100	17	77.3	0.55
+ve	0	0	5	22.7	

* *p*-value ≤ 0.05 is considered statistically significant.

**Table 6 diagnostics-13-00325-t006:** Comparison between the immunologic remission status at day 15 and the PRISM combination.

			MRD 15		*p*-Value *
			−VE	+VE	
PRISM	−VE	Count	11	0	<0.001
		% within PRISM	100.0%	0.0%	
		% within MRD 15	84.6%	0.0%	
	+VE	Count	2	7	
		% within PRISM	22.2%	77.8%	
		% within MRD 15	15.4%	100.0%	

* *p*-value ≤ 0.05 is considered statistically significant.

**Table 7 diagnostics-13-00325-t007:** The relationship between MRD D15 and death.

Death	MRD D15		*p*-Value *
	−ve (*N* = 26)	+ve (*N* = 13)	
Yes	3 (11.5%)	2 (15.4%)	1.0
No	23 (88.5%)	11 (84.6%)	

* *p*-value ≤ 0.05 is considered statistically significant.

## Data Availability

Not applicable.

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
