# Peer review of "Prognostic Role of CD200 in Acute Lymphoblastic Leukemia Patients"

_diagnostics, 2023, doi:10.3390/diagnostics13020325_

Round 1

Reviewer 1 Report

This paper proposed that CD200 can serve as a promising biomarker for the prognosis of lymphoblastic leukemia patients based on their observations in clinics. I think it is of great value for the diagnosis and treatment of this disease and of great interest to researchers in this field. Overall, the paper is well organized and its hypothesis is solidly supported by their findings. I am very glad to recommend its publication in the journal of diagnostics.

Author Response

Thanks for your supporting comments

Reviewer 2 Report

There are several points that need rephrasing/corrections

Demographic data is given in MM and Results section: should be just in the Result section.

Figure 1 is not comprehensible it its present form. Authors should either try to graphically edit it so it can elucidate a clear picture of what is being said or give a more detailed explanation. A figure should speak by itself and not need concomitant explanation.

Line 180: Sentence The best cutoff value was 60%. should be elaborated  

Line 185-187: Rephrase. „In the meantime“ is not appropriate.

Line 188-190: “In the meantime” is used ever too often and in the wrong meaning.

Discussion should focus on CD200 and not characterization of B-ALL and incidence of different chromosomal and karyotype finding and comparison to published data done on much larger cohorts as this is not relevant to studied question ( the expression pattern od CD200 on leukemic B cells)

Line 340: rephrase please

Author Response

Response to Reviewer 2 Comments

Point 1: Demographic data is given in MM and Results section: should be just in the Result section

Response 1: We edited it as required.

Point 2: Figure 1 is not comprehensible it its present form. Authors should either try to graphically edit it so it can elucidate a clear picture of what is being said or give a more detailed explanation. A figure should speak by itself and not need concomitant explanation

Response 2: Navios Cytometer software Prism is used to analyze multicolor immunofluorescence samples. Prism allows you to display percentages on all phenotypic populations in a single plot. It is software derived and can be acquired in either run time or listmode.

https://www.manualslib.com/manual/1336895/Beckman-Coulter-Navios.html?page=300&term=prism&selected=3#manual

 we provided this reference for more explanations related to PRISM figure

Point 3: Line 185-187: Rephrase. „In the meantime“ is not appropriate. Line 188-190: “In the meantime” is used ever too often and in the wrong meaning). We edited it as required.

Response 3: We edited it as required.

Point 4: Discussion should focus on CD200 and not characterization of B-ALL and incidence of different chromosomal and karyotype finding and comparison to published data done on much larger cohorts as this is not relevant to studied question ( the expression pattern od CD200 on leukemic B cells

Response 4: Our main aim is to find correlation between CD200 and ALL in MRD monitoring and to add CD200 in MRD panel and in National cancer institute one treatment protocol for patients in our study according to age (pediatric protocol based on total XV St. Jude protocol, while Adult received the Dana Farber Cancer Institute (DFCI) ALL/LBL protocol), and we didn't use different treatment protocols and no effect on remission is expected with expression of CD200 (Jassim et al.,2021)

   { Jassim Alwan M, Al-Mudallel SS. Prognostic impact of CD200 expression in pediatric B-cell acute lymphoblastic leukemia. Med J Babylon 2021;18:358-63.}

Point 5: Line 340: rephrase please

Response 5: we edited as required
